

# Genome-wide identification of the CYP82 gene family in cucumber and functional characterization of *CsCYP82D102* in regulating resistance to powdery mildew

Hongyu Wang, Pengfei Li, Yu Wang, Chunyu Chi and Guohua Ding

Harbin Normal University, Harbin, Harbin, China

## ABSTRACT

The cytochrome P450 (CYP450) gene family plays a vital role in basic metabolism, hormone signaling, and enhances plant resistance to stress. Among them, the CYP82 gene family is primarily found in dicots, and they are typically activated in response to various specific environmental stresses. Nevertheless, their roles remain considerably obscure, particularly within the context of cucumber. In the present study, 12 CYP82 subfamily genes were identified in the cucumber genome. Bioinformatics analysis included gene structure, conserved motif, cis-acting promoter element, and so on. Subcellular localization predicted that all CYP82 genes were located in the endoplasmic reticulum. The results of cis element analysis showed that CYP82s may significantly affect the response to stress, hormones, and light exposure. Expression patterns of the CYP82 genes were characterized by mining available RNA-seq data followed by qRT-PCR (quantitative real-time polymerase chain reaction) analysis. Members of CYP82 genes display specific expression profiles in different tissues, and in response to PM and abiotic stresses in this study, the role of *CsCYP82D102*, a member of the CYP82 gene family, was investigated. The upregulation of *CsCYP82D102* expression in response to powdery mildew (PM) infection and treatment with methyl jasmonate (MeJA) or salicylic acid (SA) was demonstrated. Further research found that transgenic cucumber plants overexpressing *CsCYP82D102* display heightened resistance against PM. Wild-type (WT) leaves exhibited average lesion areas of approximately 29.7% at 7 dpi upon powdery mildew inoculation. In contrast, the two independent *CsCYP82D102* overexpression lines (OE#1 and OE#3) displayed significantly reduced necrotic areas, with average lesion areas of approximately 13.4% and 5.7%. Additionally, this enhanced resistance is associated with elevated expression of genes related to the SA/MeJA signaling pathway in transgenic cucumber plants. This study provides a theoretical basis for further research on the biological functions of the P450 gene in cucumber plants.

Corresponding author
Guohua Ding,
hsddgh@hrbnu.edu.cn

## INTRODUCTION

Cytochrome P450 (P450) is a monooxygenase encoded by the B group of cytochrome superfamily proteins with heme as an auxiliary group. It is the largest and oldest family in nature and is distributed throughout all branches of the evolutionary process of life (*Guengerich, 2019*). CYP450s, named for its characteristic spectral property of an absorption peak at 450 nm when binding with CO, are characterized by a preserved heme-binding region with the sequence FxxGxRxCxG (*Bolwell, Bozak & Zimmerlin, 1994*; *Distefano et al., 2021*). P450s are involved in both basic metabolism and secondary metabolism of plants and play an essential role in enhancing plant resistance to stress and pests (*Aubert et al., 2015*; *Haduch et al., 2023*; *Minerdi, Savoi & Sabbatini, 2023*; *Pan et al., 2018*). According to the criteria of homology and phylogeny, plant P450 genes are clustered into 10 clans, including seven single families (CYP51, CYP74, CYP97, CYP710, CYP711, CYP727, CYP746) and four multiple-families (CYP71, CYP72, CYP85, and CYP86) (*Nelson et al., 2004*). Individuals within the CYP81, CYP82, and CYP93 subgroups are all part of the A-type CYP450 family, primarily engaged in producing secondary metabolites (*Guttikonda et al., 2010*). The CYP82 family, associated explicitly with the CYP71 clan, is exclusive to dicotyledonous plants.

Notably, certain members within the CYP82 group have been documented to exhibit significant induction under environmental stress conditions in various plant species, including *Nicotiana tabacum* L., *Pisum sativum L.*, *Glycine max* (L.), and *Arabidopsis* (*Whitbred & Schuler, 2000*; *Xia et al., 2023*). For instance, *AtCYP82C2* can improve jasmine-induced defense-related gene expression and jasmonic acid-induced immunoglobulin content and enhance resistance to *Botrytis cinerea* (*Liu et al., 2010*). The fact that gene SSN (*GhCYP82D*) in cotton regulates systemic cell death by regulating octadecane pathway while controlling the biosynthesis of jasmine acid to regulate resistance to *Verticillium wilt* negatively, suggest a novel metabolic branch that might regulate the JA signaling pathway (*Sun et al., 2014*). Transgenic *Nicotiana benthamiana* plants that were engineered to overexpress *GmCYP82A3* displayed robust resistance to both *Botrytis cinerea* and *Phytophthora parasitica*, along with increased tolerance to salinity and drought-induced stresses. Moreover, these genetically modified plants exhibited reduced sensitivity to JA, and this heightened resistance was accompanied by an upregulation of genes related to the JA/ET signaling pathway (*Yan et al., 2016*). The findings indicate a significant role of the CYP82 subfamily in plant stress resistance. However, additional analysis is required to elucidate the specific functions of the CYP82 subfamily in cucumber.

The discovery of key genes in the P450 family is helpful for the search on stress tolerance in various species. Cucumbers (*Cucumis sativus* L.) are one of the main vegetable crops in China and have medical, therapeutic, and nutritional values (*Abbas et al., 2023*; *Akram et al., 2023*). Nonetheless, limited research exists on cucumber P450 genes. In our previous research, SA transcriptome analysis was performed on the cucumber. A set of differentially expressed genes (differential expression genes, DEGs) were acquired, and among them are genes that fall within the CYP82 subfamily. The cytochrome CYP82 subfamily in the

cucumber genome was identified using bioinformatics analysis to fully understand the gene structure and protein function. Gene structure, conserved motifs, cis-acting promoter elements, evolutionary profiles, collinearity, and subcellular localization were analyzed. CsCYP82 genes were responsible for cucumber development, light response, hormonal response, and responding to biotic and abiotic stresses. Analysis of the phylogenetic tree and motifs revealed that the CYP82 proteins are evolutionarily conserved. The findings are intended to provide a theoretical basis for further exploration of the biological function of the cucumber P450 protein. Furthermore, we scrutinized the expression profiles of all CYP82 subfamily genes within cucumbers, dissecting their expression patterns and isolating a promising candidate, *CsCYP82D102*. This study conducted a comprehensive analysis of *CsCYP82D102*, dissecting its involvement in responses to both biotic and abiotic stresses. Our results demonstrate that overexpressing *CsCYP82D102* in cucumber enhances resistance to pathogens like PM. Furthermore, our research unveils a significant modification in the SA/MeJA signaling pathway within the overexpressing plants, providing valuable insights into the mechanisms underpinning partial resistance in cucumbers. The findings of this study collectively contribute to a deeper understanding of P450 enzymes in plants and their pivotal roles in mediating stress responses, particularly in the context of cucumber.

## MATERIALS AND METHODS

### Plant materials and stress treatment

Portions of this text were previously published as part of a preprint (https://doi.org/10.21203/rs.3.rs-1196334/v1) (*Wang et al., 2021*). Cucumber 9930 cultivated in North China was selected as plant material (donated by Prof. Huang Sanwen, Institute of Vegetable Research, Chinese Academy of Agricultural Sciences). The 9930 seeds were placed in a clean culture dish with wet filter paper. After the cotyledons are opened, they were moved to the soil (vermiculite: nutrient soil = 1: 1) in the greenhouse at 22 ± 4 °C with a photoperiod of 16 h light and 8 h dark. All materials, including roots, stems, and leaves were immediately frozen in liquid nitrogen and stored at −80 °C until total RNA isolation.

Four-week-old cucumber plants were exposed to abiotic stresses and defense-modulating plant hormones. Experimental solutions were prepared using methanol as a solvent for 100 μM JA, while distilled water was used as a solvent for 100 μM salicylic acid (SA), 100 μM abscisic acid (ABA), 100 μM ethylene (ETH), 250 mM NaCl, and 20% polyethylene glycol (PEG 6000). At specified time intervals (0, 3, 12, 24, and 48 h) after treatment, plantlets from each treatment group and control samples were collected. The collected samples were swiftly frozen in liquid nitrogen and preserved at −80 °C.

### Identification of CYP82 subfamily members in *Cucumis sativus* L.

The CYP82 subfamily was identified and analyzed based on the complete cucumber genome. Pertinent genomic and protein data for cucumber were retrieved from the database (http://cucurbitgenomics.org/). The Hidden Markov model for the P450 structural domain (PF00067) was acquired from the Pfam database (http://pfam.xfam.org/) in this study. Subsequently, the Simple HMM Search program within the TB tools software

was employed to search the cucumber genome and predict the IDs of the CYP82 gene family. These sequences were further validated using the Protein BLAST function in the NCBI database (https://blast.ncbi.nlm.nih.gov/Blast.cgi). The cucumber (Chinese Long v2) genome, CDS, transcripts, polypeptides, and a 2,000 bp segment upstream of the promoter region, was retrieved from the database, spanning from the translation initiation site (ATG). ClustalX was utilized to analyze the amino acid sequences of the target CYP82 sequences. Additionally, the neighbor-joining (NJ) method parameter in MEGA7.0 was employed, utilizing pairwise deletion and 1,000 bootstrap replicates for multiple alignments in constructing a phylogenetic tree among the CYP82 family.

Next, to identify all potential P450 genes in the cucumber genome, a BLAST search against the cucumber genome was conducted using 245 AtCYP450s genome sequences downloaded from Cytochrome P450 database (https://drnelson.uthsc.edu/plants/). Based on the 230 cucumber P450 gene sequences obtained from the P450 homepage, and by comparing them with the downloaded *Arabidopsis* sequences to find a sequence with higher consistency, each gene is classified and named according to international standards.

## Conserved domain, motif identification, and gene structure analysis

The conserved motif of cucumber P450 was analyzed using the online tool MEME (http://meme-suite.org/tools/meme). The maximum number of motifs was 10, and the optimized motif width was 10–100 amino acid residues. The rest of the parameters are default (*Hu et al., 2015*). The location information of the gene and the conservative area was confirmed and then plotted using the Perl SVG package.

## Phylogenetic analysis

The full-length amino acid sequences of gene family members derived from closely related species were used for phylogenetic analysis. The MEGA program 7.0 performed multiple sequence alignments of the obtained genes. The phylogenetic tree was generated using the Neighbor-Joining (NJ) technique in MEGA and a bootstrap test conducted 1,000 times (*Kumar, Stecher & Tamura, 2016*).

## Gene replication

The potential replication genes were identified using the duplicate MCScanX software's gene classifier program (*Wang et al., 2012*). The genome's all coding gene protein sequences were aligned consuming blastP, and the alignment consequences were used as input files for the MCScanX Software for reproduction gene projection. The gene was identified as a reproduction gene pursuant to e-value $< 1e^{-5}$ or e-value $< 1e^{10}$.

## Promoter cis-element analysis

Intercepting the upstream sequence (2.0kb) of each CsCYP450 gene annotation file in Cucurbitaceae database (http://cucurbitgenomics.org/), you then proceeded to scan the cis-acting elements from those sequences. Cis-acting elements prevalent in genes were identified, including CAAT-box, TATC-box, TATA-box, and cis-acting elements with specific functions were only shown (*Lv et al., 2020*; *Trapnell et al., 2010*; *Wang et al., 2020*).

## The expression analysis of CYP82 genes

Cucumber transcriptome data (PRJNA80169) was acquired from the Cucurbit Expression Atlas within the Cucurbit Genomics Database (CuGenDB). A detailed analysis of the expression profiles of 12 CYP82 genes was performed using the FeatureCounts R package, and these patterns were subsequently visualized as heatmaps utilizing TBtools (*Yang et al., 2022*). Concurrently, heatmaps representing the expression patterns of CYP82 genes in response to PM treatment were generated using cucumber transcriptome data (PRJNA321023) (*Yang et al., 2022*).

## Quantitative reverse transcription PCR (qRT-PCR) expression analysis

According to the results of combining CYP82 family with RNA-seq analysis, 10 candidate genes were selected for qRT-PCR analysis. Primers for qRT-PCR were designed by using Primer Premier 5.0 software (Table S1). The cucumber Actin gene was used as a control for normalization between samples. Total RNA was extracted from each samples using RNA pure Plant Kit (Cwbio, Beijing, China) according to the manufacturer's instructions. First-strand cDNA synthesis was performed using an oligo (dT) primer and 2 g of total RNA in a 20-L reaction volume, according to the manufacture's instruction for the Super RT cDNA Synthesis Kit (Cwbio, Beijing, China). The integrity of RNA was assessed by 1% agarose gel electrophoresis, qRT-PCR was carried out using a Ul'tra SYBR Mixture (WithRox) kit (Cwbio, Beijing, China) with a Fast Real-Time PCR System (Applied Biosystems 7500, USA). The PCR mixture consisted of 1 μL cDNA template, 1 μL 10 μM forward primer, 1 μL 10 μM reverse primer, 12.5 μL Ul'tra SYBR Mixture, and 9.5 μL RNase-free water. The PCR verification pictures are shown in Fig. S1. The recommended conditions for PCR were used as follows: 95 °C for 10 min, followed by 40 cycles of 95 °C for 30 s, 58 °C for 30 s, and 72 °C for 30 s. All reactions were performed in triplicate for each sample. The relative expression analysis of each gene were calculated using the $2^{-\Delta\Delta CT}$ method (*Duan et al., 2020*).

## Subcellular localization of CYP82D102

The site Plant-mPLoc 2.0 (http://www.csbio.sjtu.edu.cn/bioinf/plant-multi/), was used to make predictions regarding subcellular localization (*Ai et al., 2020*). This bioinformatic investigation disclosed that *CsCYP82D102* predominantly localizes within the endoplasmic reticulum (ER). For subsequent experimental validation, the coding sequence (CDS) of *CsCYP82D102* was successfully cloned into the pCAMBIA1302-GFP vector, omitting stop codons, precisely at the *Nco*I and *Bgl*I. sites. The GFP-CYP82D102 fusion construct was introduced into Arabidopsis protoplast cells. In this assay, the Cauliflower Mosaic Virus (CaMV) 35S and pHB-GFP vectors were employed as controls, with the fusion construct serving as the experimental treatment. The protoplast preparation method is detailed in Shen et al.'s work (*Shen et al., 2014*). The samples were then observed using a Leica TCS SP5-II confocal fluorescence microscope. The primers employed for vector construction can be located in Table S1.

## Plasmid construction and generation of transgenic plants

To generate *CsCYP82D102* overexpressing (OE) plants, the complete cDNA fragment of *CsCYP82D102* (1617 bp) was cloned into the binary expression vector pBI121, which is driven by a CaMV 35S promoter. *Agrobacterium tumefaciens* strain LBA4404 was employed for plant transformation. The desired fragment was amplified through the Golden Gate seamless cloning technique for RNAi analysis (*Yan et al., 2012*). It was then inserted into the vector and delivered into cucumber using LBA4404. The procedures for generating OE and RNAi plants followed a similar protocol, with modifications based on the method described by *Liu et al. (2023)* for genetic transformation in cucumber. These transgenic lines were planted in a greenhouse self-pollination to obtain T1 generation seeds.

## Physiological index detection of the transgenic cucumber

Enzyme activities associated with reactive oxygen species (ROS), including superoxide dismutase (SOD), peroxidase (POD), and catalase (CAT), were determined following established protocols. Additionally, malondialdehyde (MDA) content was quantified using previously described methods. Marker gene expression in the SA and MeJA signaling pathways was analyzed using qRT-PCR. Detection of superoxide anion ($O_2.^-$) and hydrogen peroxide ($H_2O_2$) accumulation, as well as cell death in cucumber leaves, was performed *via* histochemical staining with 3,3′-diaminobenzidine (DAB) and nitroblue tetrazolium (NBT) respectively (*Daudi & O'Brien, 2012*; *Jambunathan, 2010*).

## Statistical analyses

Three biological replicates and three technical replicates were executed to uphold the reliability of our findings. The gathered data for comprehensive analysis employing data processing software such as Origin 8.0 and DPS 9.5. Subsequently, the results were presented as the mean ± standard deviation (SD). Noteworthy distinctions between the treatment and control groups were assessed using Student's t-tests, with statistical significance set at $P < 0.05$. Additionally, a higher level of significance ($P < 0.001$) was considered exceptionally significant. Moreover, the amassed data underwent further scrutiny through analysis of variance (ANOVA).

# RESULTS

## Identification and physicochemical properties analysis of CYP82 protein in *Cucumis sativus* L.

After removing the redundant genes, 12 CYP82 family members were identified in the cucumber genome (Table S2). The protein sequences encoded by these genes were further confirmed by NCBI and SMART screening to contain four conserved motifs, namely the FxxGxRxCxG heme-binding protein domain, I-helix, K-helix, and PERF/W motif. The size of cucumber P450 protein was between 317 and 1,056 aa, the theoretical isoelectric point (pI) of the encoding protein was 6.85–9.52, and the molecular weight was between 36,767.9 and 65,839.3 kDa. The subcellular localization predicted that all the

CYP82 genes were located in the endoplasmic reticulum. These results suggest that *CsCYP82* plays a significant role in cucumber's cellular structure and function.

## Analysis of gene structure, conserved domain, and motif of CYP82 subfamily in *Cucumis sativus* L.

Based on the alignment results of the coding sequence to the genomic sequence of the cucumber CYP82 gene family, the exon numbers of these CYP82 genes varied from 1 to 10, revealing the structural diversity of the P450 gene family (Figs. 1A–1C). Of the cucumber CYP82 gene family, the Csa3G852560 gene contained the most exons. Further analysis showed that most of the same-family members had similar distribution characteristics regarding exon length and number. Conserved motif analysis of P450 in cucumbers is shown in Fig. 1C. A total of 10 conserved motifs were identified and named motif 1 to motif 10, respectively (Table S3). Six motifs were found in more than 90% of the genes. Except for Csa3G853140 and Csa3G852619, all CYP82 gene members contained motif 4: FGAGRRICPG (FxxGxRxCxG), which was annotated as the core functional domain (the heme-binding domain). The heme-binding domain contains critical cysteine residues, which are axial ligands of heme. Motif 3 was annotated as a helix K domain (EXXR) that plays an important role in P450 protein folding. Motif 3 was annotated as an I-helix conserved domain involved in oxygen binding and oxygenation. This result shows that motif 3 is highly conserved during cucumber development. Motif 6 was annotated as a PERF domain and motif 7 was annotated as a PKG motif. Members of the same gene cluster shared similar conserved domains, whereas different gene clusters had specific conserved domains. In conclusion, the highly conserved motif and similar gene structure of CYP82 in the same family further support the phylogenetic analysis's close evolutionary relationship and reliability.

## Analysis of replication events of cucumber CYP82 subfamily

The process of gene duplication stands as a pivotal evolutionary mechanism, fostering genetic diversity and the emergence of novel functions, thereby significantly contributing to adaptation and speciation. Gene duplications are categorized as segmental, tandem, proximal, singleton, and dispersed. Among these categories, the prevalence of tandem duplication accounted for 38.6% (76 instances), while fragment duplication represented 19.3% (38 instances), proximal duplication constituted 9.1% (18 instances), decentralized duplication encompassed 33% (65 instances), and no single-copy duplication was observed (refer to Fig. 1D). The findings underscored that tandem replication acts as the primary driving force behind the amplification of the CsCYP450 gene family. This particular mode of replication is crucial for sustaining an extensive gene family, enabling rapid expansion or contraction in response to environmental variations, thereby enhancing genetic complexity, especially under favorable conditions (*Yu et al., 2017*). All members of the CYP82 subfamily are located on chromosome 3 of cucumber. Except for Csa3G852560, the others are tandem replication. Moreover, the chromosomal distribution pattern of CsCYP450 provides strong indications that the proliferation of CsCYP450 genes in cucumbers is significantly influenced by tandem replication.

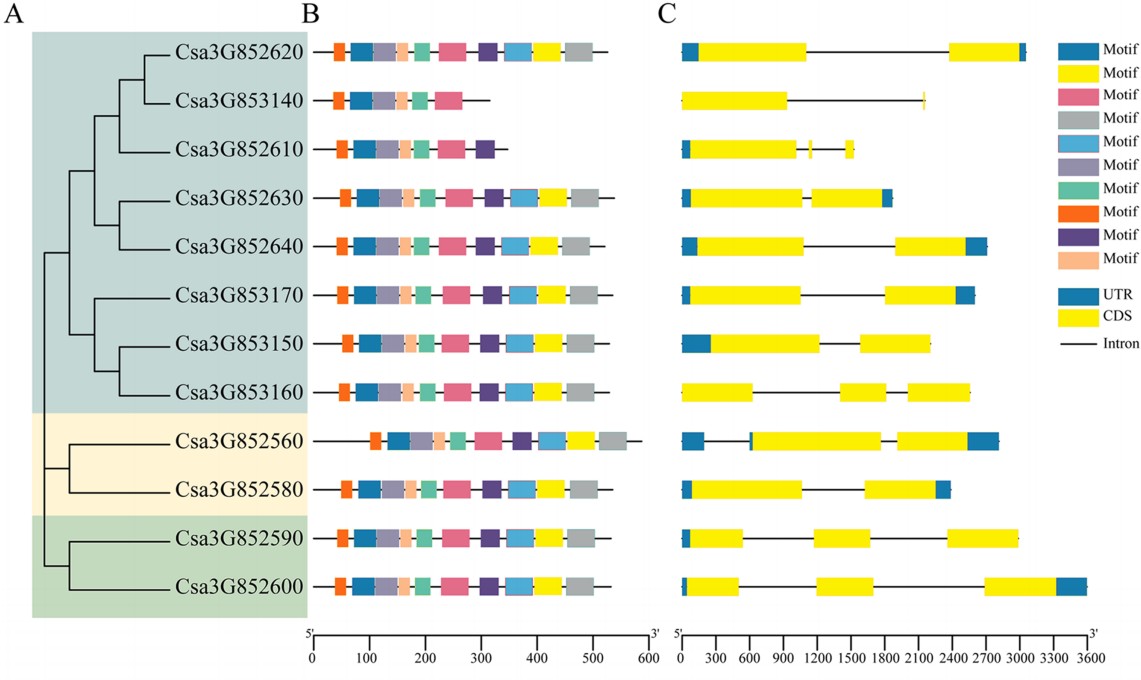

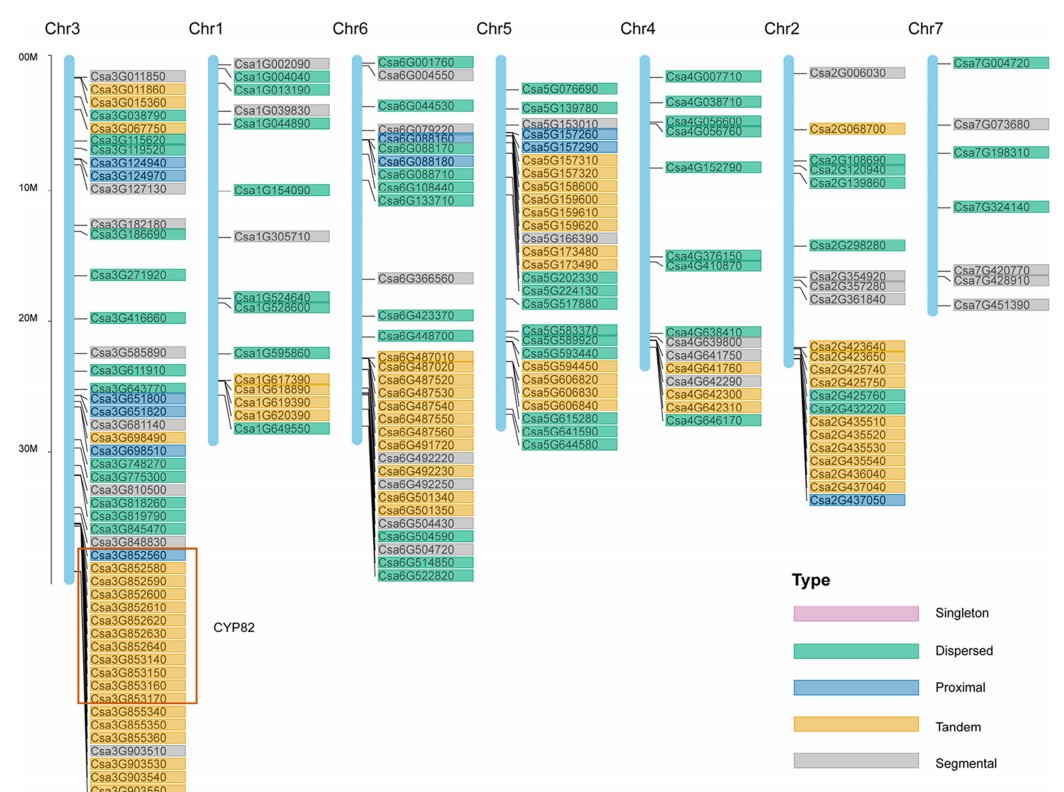

**Figure 1 Gene structure analysis and replication gene types of cucumber CYP82 members.** (A) Phylogenetic tree was constructed using MEGA 7 for the full-length amino acid sequences of 12 CYP82 proteins, and bootstrap was set to 1,000 times. (B) Exon/intron analysis. Yellow squares represent exons, black lines represent introns, and green squares represent upstream/downstream regions of the gene. (C) Analysis of the conserved motifs of CYP82 proteins. CYP82 protein motif, numbered 1–10, are shown in squares of different colors. Box length corresponds to motif length. The sequence information of each motif is shown in Table S3. (D) Distribution map of replication gene types of P450 protein family on chromosomes in cucumber; arranged by chromosome length, with scale on the left of the third stained chromosome.

## Evolutionary analysis of related species

The genetic relationships within the cucumber P450 gene family were examined, and a phylogenetic tree was generated using the results from the multiple sequence alignment. Subsequent to the phylogenetic analysis, the 165 P450 genes were categorized into two major groups: the A-type and non-A-type P450 gene families (Fig. S2). The A-type P450 gene family was found within the CYP71 clan, while the non-A-type P450 gene family was subdivided into seven distinct clans, denoted as CYP85, CYP86, CYP72, CYP97, CYP711, CYP51, and CYP710. The phylogenetic relationships of the CYP82 protein family were explored using full-length amino acid sequences from cucumber, garden pea, opium poppy, sweet basil, soybean, and Arabidopsis (Fig. 2). It was observed that the cucumber CYP82 is in close proximity to soybean, garden pea, peppermint, and sweet basil in terms of phylogenetic relationships.

The P450 genes of cucumber were compared with CYP82 subfamilies in Arabidopsis, tomato, soybean, maize, rice, poplar, grape, and moss (Table S4). CYP82 has evolved in the cucumber genome compared with *Arabidopsis*. There are 34 CYP82 family members in grapes and 24 in soybeans. Some studies have shown that CYP736, CYP83, and CYP82 are very similar, and they strongly induce *Phytophthora sojae* to infect cucumber hypocotyls (*Guttikonda et al., 2010*). Nevertheless, in the genomes of cucumber, soybean, and grape, CYP82 exhibits a greater number of P450 members, whereas it is not present in the genomes of rice and moss.

## Collinearity analysis

Collinearity was originally used to describe the locations of genes on the same chromosome. It now refers to the conservation of gene types and relative order in different species derived from the same ancestral type. Collinearity analysis can identify linear homologous genes among species, annotate protein-coding genes, and discover evolutionary events. The collinearity of species was constructed using McScanX and plotted using the Circos software (Fig. S3). The CYP82 subfamily genes of cucumber, melon, and Arabidopsis genomes were jointly analyzed to study their collinear genetic relationships Collinearity analysis showed that zero pairs of collinear genes were identified between the genomes of Arabidopsis and cucumber. There were three collinear gene pairs between cucumber and melon (Fig. S2). These results suggest that the functional differentiation of these genes may have occurred in cucumbers and melons during evolution.

## Analysis of cis-acting elements of *CsCYP82* promoter

To better understand the potential regulatory mechanism of *CsCYP82* during cucumber growth and development, the cis-regulatory element in the promoter regions of *CsCYP82* was identified in this study. The upstream sequence (2.0 kb) of all CsCYP82 translation initiation sites was scanned and the potential role of CsCYP82 expression elements was predicted using the Plant Care tool (Fig. 3).

In addition to the TATA-box, CAT-box, and other specific elements, the *CsCYP82* promoter contains various cis-regulatory elements related to light signal response, tissue

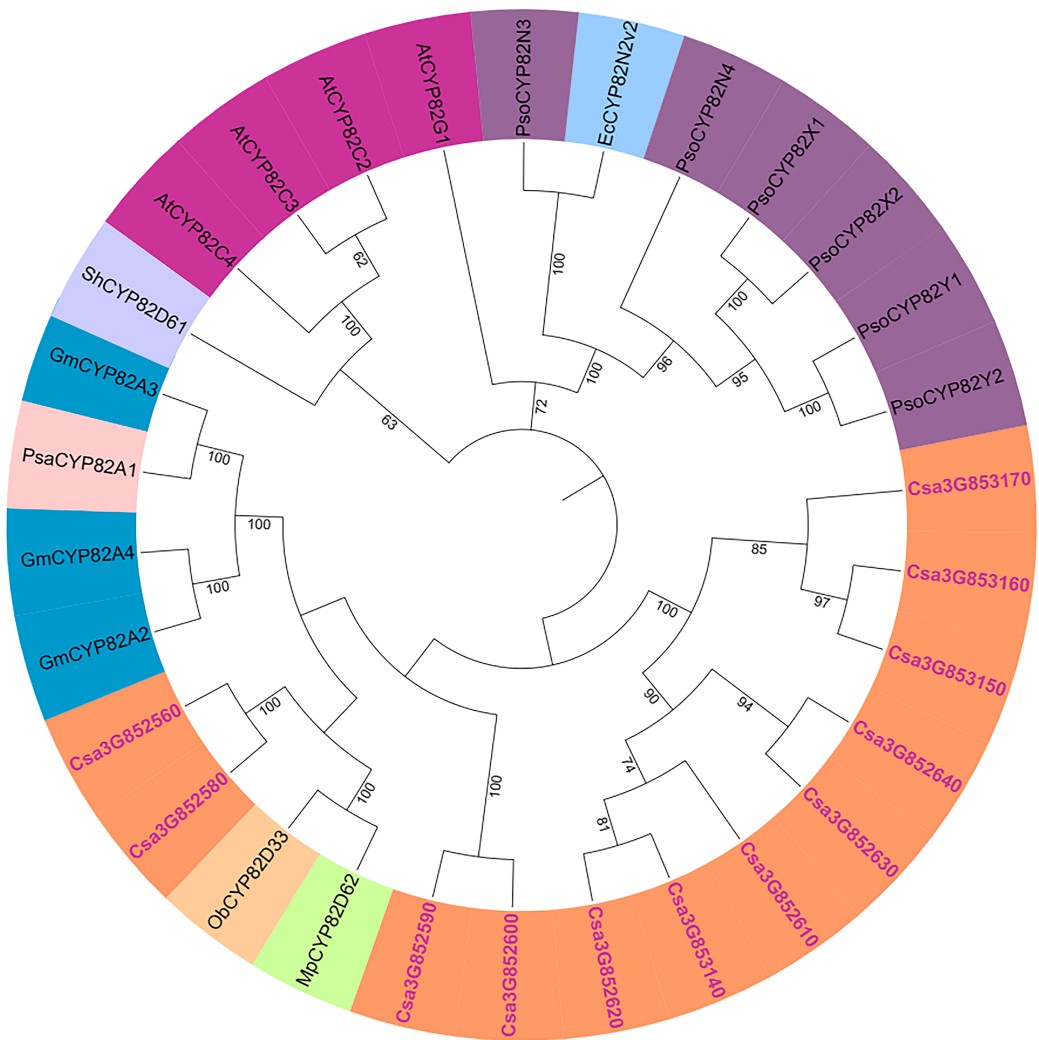

**Figure 2 Phylogenetic tree of CYP82 gene family in a variety of plants.** Including *Cucumis sativus* (Cs), *Arabidopsis thaliana* (At), *Papaver somniferum* L (Pso), *Sinopodophyllum hexandrum* (Sh), *Eschscholzia californica* (Ec), *Mentha piperita* L. (Mp), *Ocimum basilicum* L. (Ob), *Glycine max* (L.) (Gm), and *Pisum sativum* (Psa).

and organ development, hormone response, defense, and stress. Many regulatory elements related to the light response were detected in the *CsCYP82* gene, such as ACE, ATCT-motif, and CAG-motif. Most genes contain at least two box4 elements, and there are six Box4 elements in the promoter region of the Csa3G852640 gene, suggesting that some DNA modules may be involved in the light response.

Among the elements related to plant growth and development, nine specific regulatory elements were identified, including those related to cell differentiation (CAT-box), endosperm expression (GCN4-motif), and zein metabolism regulation ($O_2^-$ site). Csa3G852580, Csa3G852560, and other genes were identified as cis-acting regulatory elements involved in regulating circadian rhythm, which means that their function may be affected by day length.

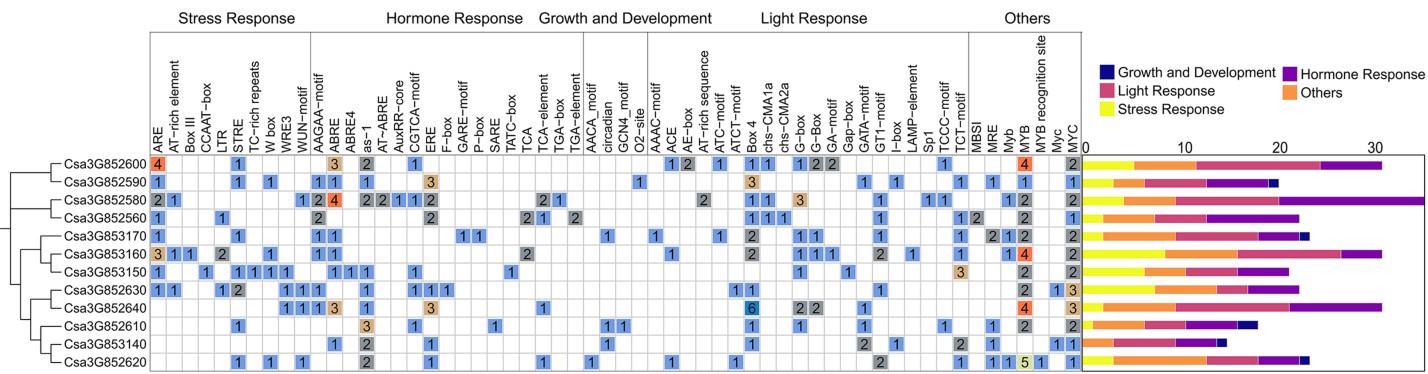

**Figure 3 Analysis of cis element of CYP82 member promoter in cucumber.** Potential cis-elements in a 2.0 kb 5′flanking region upstream from the start codon of each CsCYP450. The number of each cis-element was shown, and the back-color changes from blue to red as the number increase. All cis-regulatory elements were classified into four groups, including light, hormones, growth, biotic and abiotic.

Among the elements related to plant hormones, 11 hormone response regulatory factors were identified, which were related to abscisic acid (*Jayakannan et al., 2015*; *Martín et al., 2011*), GA (GARE-motif, P-box, TATC-box), IAA (AuxRR-core, TGA-box, TGA-element), SA (TCA-element, SARE), and MeJA (CGTCA-motif, TGACG-motif). Among these, the ABA-related response elements are widely distributed. The promoter regions represented by the Csa3G852580 gene was predicted to contain four ABRE elements, which might be involved in the ABA signaling pathway, and SA and GA elements were enriched in most CYP82 promoters, indicating that SA and GA may induce them.

Among the biotic and abiotic stress-related elements, ARE elements were found in most gene promoter regions. Four ARE elements were found in Csa3G852600, and three ARE elements were found in Csa3G853160. ARE is necessary for cis-acting element-based anaerobic induction and may directly affect the antioxidant capacity of the gene. ERE elements have been found in the promoter regions of some genes, such as Csa3G852590 and Csa3G852640. In addition, a few genes were rich in WUN-motif, LTR, and DRE response elements, suggesting that they may be interested in wound response, low temperature, and osmotic stress response mechanisms. These results show that CYP82s may significantly affect the response to stress, hormones, and light exposure.

## Expression profiles of CYP82 genes in *Cucumis sativus* L.

P450 genes assume vital functions within a wide array of biological pathways, with their distinct expression patterns in specific tissues intricately connected to the diverse roles these genes perform in various tissue types. A previous study comprehensively examined the tissue-specific expression of CYP82 genes in cucumber, encompassing roots, stems, leaves, male and female flowers, ovaries, and tendrils through an RNA-seq data analysis. Remarkably, Csa3G852610, Csa3G852560, Csa3G852600, and Csa3G852580 demonstrated consistent expression across all examined tissues, underscoring their significant contributions to cucumber development (Fig. 4A). Furthermore, Csa3G852590 and Csa3G852640 exhibited heightened transcriptional levels in tendrils compared to
other tissues, whereas Csa3G852630 displayed relatively elevated expression in the roots. These distinctions suggest diverse roles for CYP82 genes in the development of cucumber.

Furthermore, qRT–PCR analysis was conducted on six distinct CYP82 genes within six different tissue types, namely female flowers, male flowers, leaves, roots, stems, and tendrils (refer to Fig. 4B). The qRT–PCR results exhibited consistent trends when compared to the transcriptome data. Notably, Csa3G852600, Csa3G852610, and Csa3G852590 displayed notably higher expression levels in tendrils in comparison to the other examined tissues. Conversely, the expression of Csa3G852580 in leaves, female flowers, and stems exceeded that observed in roots and tendrils. Csa3G853140, on the other hand, showed pronounced expression in roots but the lowest expression in stems.

In order to investigate the involvement of CYP82 genes in responding to biotic stress, the TPM values of CYP82 genes were collected from the RNA-seq data following powdery mildew (PM) infection and subsequently constructed heatmaps (Fig. 4C). During the infection of *Sphaerotheca fuliginea* (PM treatment), the expression profiles of three CYP82 genes exhibited significant alterations in cucumber varieties. Specifically, Csa3G852610, Csa3G852630, and Csa3G853160 demonstrated increased expression levels in response to PM treatment in both susceptible and resistant varieties, implying their pivotal roles in conferring powdery mildew resistance. These findings suggest a significant role for CYP82 genes in cucumber disease resistance during infection.

Furthermore, the expression patterns induced by PM inoculation of the selected genes were also assessed (refer to Fig. 4D). Following PM inoculation, it became evident that the expression patterns in Csa3G852630 and Csa3G853160 closely resembled each other, with the induced expression of these genes peaking at 24 h post-inoculation. In contrast, the expression of Csa3G852640 remained unaltered in response to PM. These findings indicate that Csa3G853160 exhibits a more rapid response to PM in cucumber when compared to other CYP82 genes. These results provide evidence that Csa3G853160, Csa3G852610, and Csa3G853160 may play a role in conferring resistance to PM.

## Profiles of *CsCYP82D102* genes expression under different treatments

Because *CsCYP82D102* (GenBank accession number: XP_004147821.1) gene reacts strongly and quickly in the face of SA treatment, it was selected for further experiment. Expression analyses were conducted in different plant tissues to investigate the function of *CsCYP82D102* in cucumbers. The expression levels of *CsCYP82D102* were assessed in cucumber leaves, stems, flowers, and roots. The findings revealed that *CsCYP82D102* exhibited the highest expression in the roots, followed by the leaves, while the expression in the stems was the lowest. Upon treatment with $GA_3$, ETH, and ABA, the expression of *CsCYP82D102* decreased to varying extents compared to control plants. Conversely, exogenous SA and MeJA treatments resulted in an upregulation of *CsCYP82D102* expression. Specifically, SA treatment remarkably increased the relative expression level of *CsCYP82D102* by more than 1633-fold at 48 h. MeJA treatment slightly elevated the relative expression of *CsCYP82D102* by more than 238-fold at 24 h (Figs. 5A–5H). These findings suggest that *CsCYP82D102* may participate in the SA and MeJA signaling pathways.

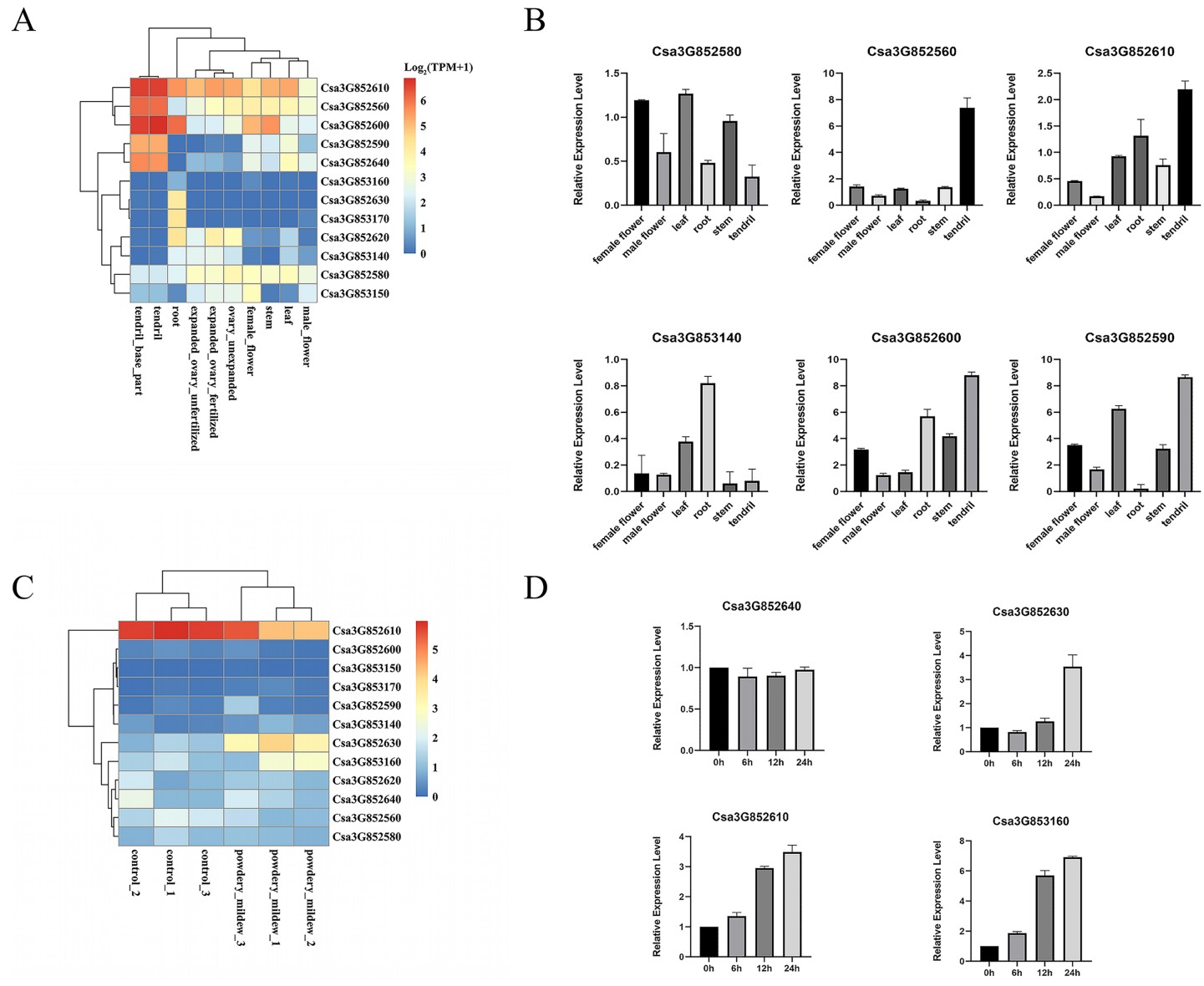

**Figure 4 Illustrates the expression patterns of CYP genes in cucumber.** (A) The expression profiles of CYP82 genes across various cucumber tissues. Transcript abundance of CYP82 genes was calculated by the TPM (Transcripts Per Million) values with FeatureCounts R package. The expression values (TPM+1) were used to plot heatmap after log2 conversion. The color scale represents relative expression levels from low (blue colored) to high (red colored). (B) A qRT–PCR analysis of CYP82 genes in different tissues is presented, with the values representing the means ± standard deviation of three replicates. For qRT-PCR, the *CsActin* gene served as an internal control. (C) The expression profiles of CYP82 genes under powdery mildew treatment; (D) the expression patterns of CYP82 genes in response to powdery mildew in cucumber.

## Overexpression of *CsCYP82D102* in cucumber enhances resistance to PM

To investigate the function of the P450 family gene, *CsCYP82D102*, in cucumbers, transgenic plants overexpressing this gene under the control of the 35S promoter were generated. Two independent T1 generations were selected for functional characterization. Genomic PCR analysis confirmed the integration of the transgene, while expression

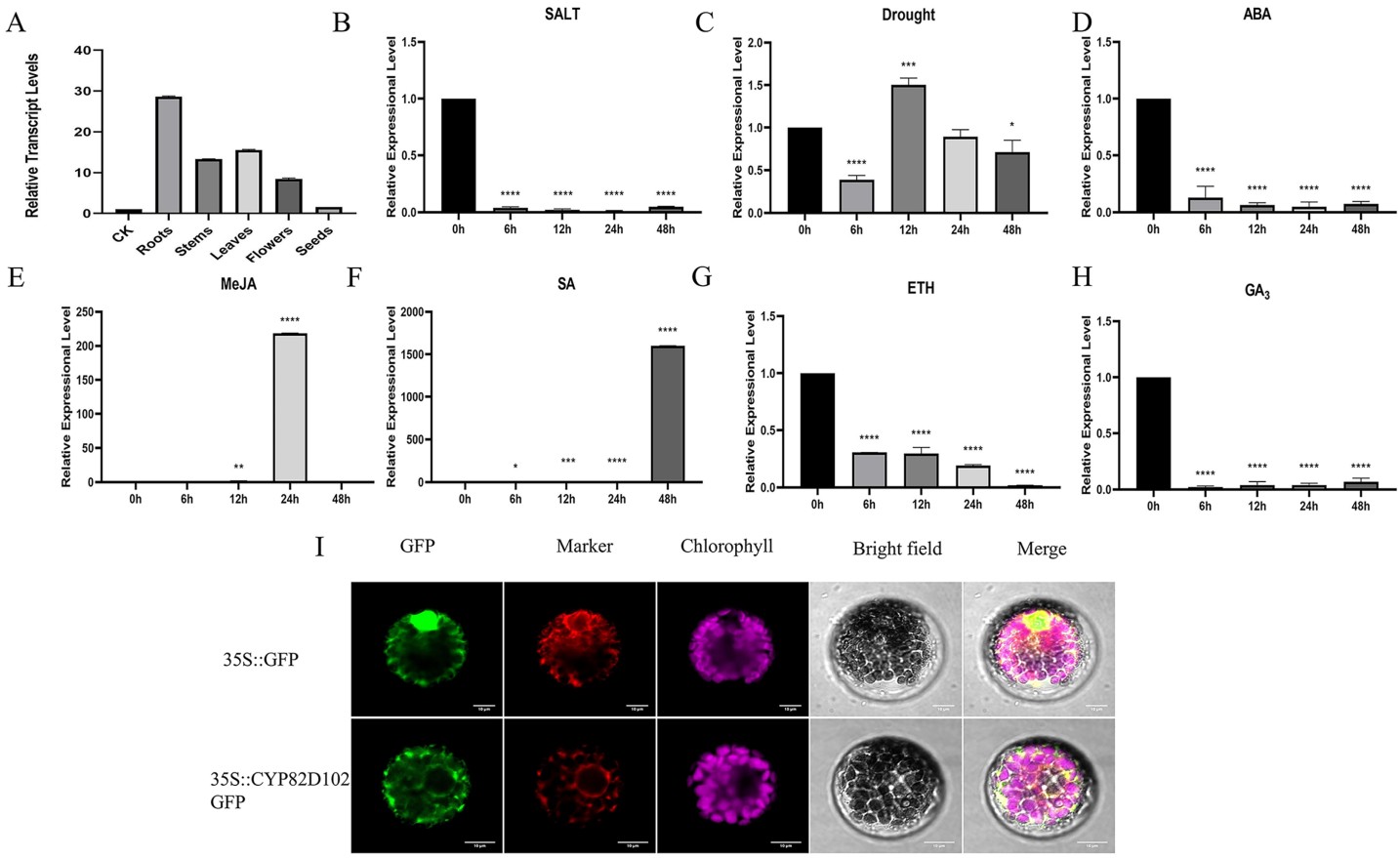

**Figure 5 The expression profiles of *CsCYP82D102* response to various abiotic stress and phytohormones.** (A–H) The relative expression level was normalized to cucumber *CsActin*. Mean and standard deviation (SD) were calculated from three independent biological replicates. (I) The subcellular localization of CYP82D102. Progressing from the left to the right, the visuals showcase the GFP's green fluorescence, spontaneous emission of chloroplast in red fluorescence, the bright field, and the merged microscope images (scale bars: 10 μm). The asterisks at the top of the columns indicates significant differences ($**P < 0.01$, $***P < 0.001$, $****P < 0.0001$, One-way ANOVA).

analysis verified its transcription (Figs. S4A–S4C). Subcellular localization showed that CYP82D102 was located in the endoplasmic reticulum, consistent with the prediction (Fig. 5I). Wild-type(WT) leaves exhibited average lesion areas of approximately 29.7% at 7 dpi upon powdery mildew inoculation. In contrast, the two independent *CsCYP82D102* overexpression lines displayed significantly reduced necrotic areas, with average lesion areas of approximately 13.4% and 5.7%. At 15 dpi, the lesion area of WT leave was 70.8%. The lesion area of OE#1 and OE#3 was 32.2% and 27% respectively (Fig. 6B). Subsequently, the levels of reactive oxygen species (ROS) were evaluated in transgenic plants. Notably, WT plant leaves exhibited prominent staining for $H_2O_2$ and $O_2^-$, whereas leaves of *CsCYP82D102* overexpressing plants displayed relatively weaker DAB and NBT staining (Fig. 6C). Furthermore, under PM stress conditions, all transgenic plants showed enhanced activity of POD, CAT, and SOD compared to WT plants (Fig. 6D). In silent lines, the difference was not significant (Fig. S5). These results indicate that *CsCYP82D102* enhances resistance against powdery mildew.

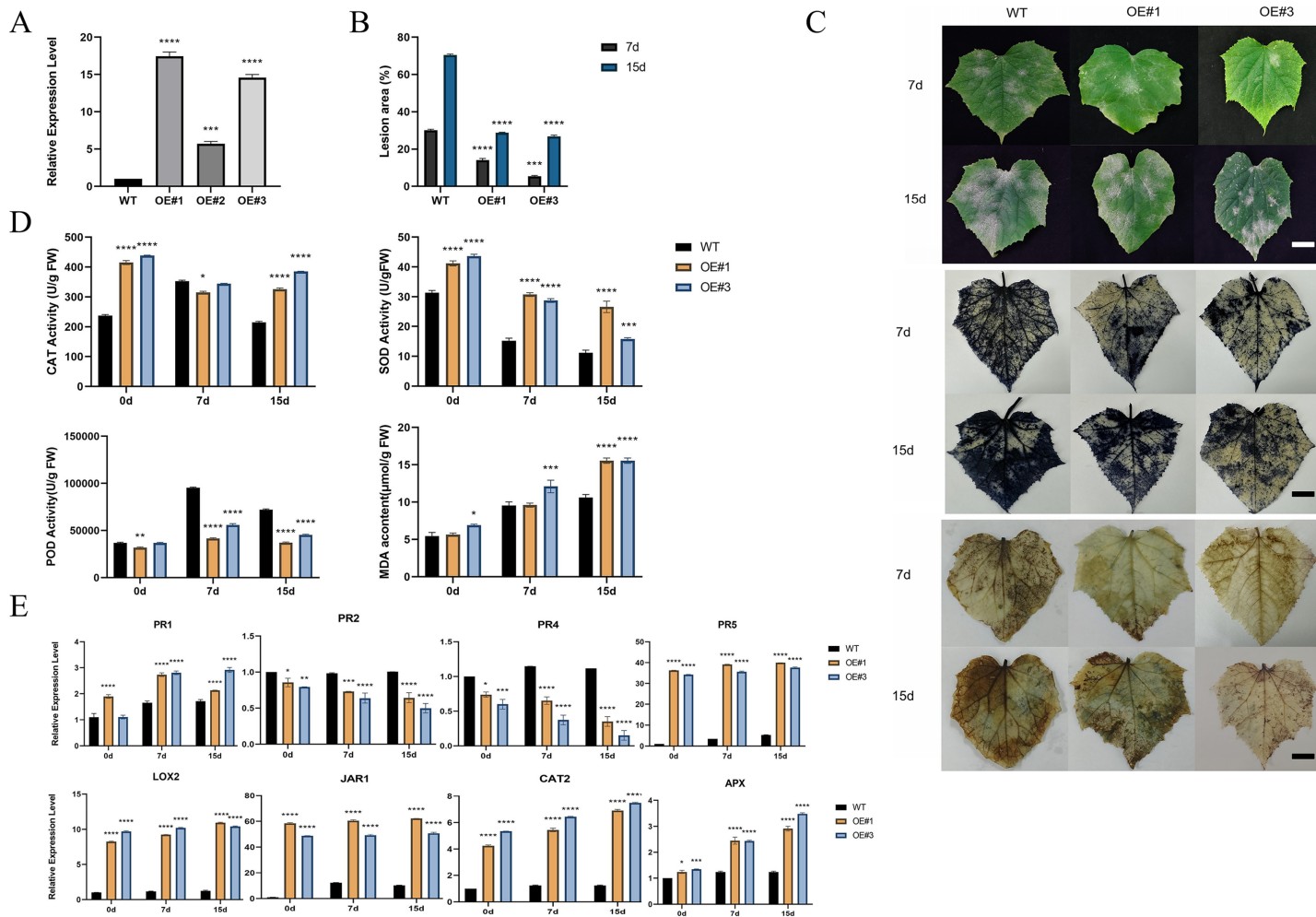

**Figure 6 Enhanced resistance to powdery mildew of transgenic cucumber lines.** (A) Expression levels of the overexpressed *CsCYP82D102* gene in transgenic plants. (B) Lesion dimensions on inoculated leaves were assessed by measuring diameters at 7 and 15 dpi, and subsequently, the lesion area was computed. Consistent findings were noted in a minimum of three separate repetitions. Standard deviation is represented by the error bars. (C) The phenotypic characteristics and DAB (3,3′-diaminobenzidine), NBT (nitro blue tetrazolium) staining of cucumber leaves from wildtype (WT) and *CsCYP82D102* overexpression lines (OE#1, OE#3) following powdery mildew (PM) inoculation were examined. Transgenic and WT leaves were infected with PM mycelia grown on PDA medium. Images were captured at 7- and 15-days post-inoculation (dpi). Scale bar = 50 mm. (D) Levels of MDA (malondialdehyde) content, along with the activities of SOD (superoxide dismutase), POD (peroxidase), and CAT (catalase) in the leaves of transgenic plants of the T1 generation were assessed both before and after inoculation with PM, respectively. (E) Expression patterns of defense marker genes as well as key genes involved in the JA/SA signaling pathway in transgenic cucumber plants. RNA samples were extracted from detached leaves of both wildtype (WT) and *CsCYP82D102* overexpression plants after powdery mildew (PM) treatment. qRT-PCR was performed using gene-specific primers, and expression levels were normalized to *CsActin*. The data presented represent the average of three independent replicates, with error bars indicating the standard deviation (SD). Significant differences between WT and transgenic plants are denoted by asterisks (**$P < 0.01$, ***$P < 0.001$, ****$P < 0.0001$, One-way ANOVA).

## *CsCYP82D102* is involved in the SA/MeJA signaling pathway

Considering the changes in resistance levels to powdery mildew (PM) infection, the expression pattern of defense-related marker genes in the defense cascade was examined in *CsCYP82D102* overexpression plants. Insights into potential regulatory pathways were aimed to be gained (Fig. 6E). Four widely recognized pathogenesis-related (PR) genes were selected for comparative analysis. Both the SA marker genes, *PR1* (pathogenesis-related

protein 1) and *PR2* (pathogenesis-related protein 2), exhibited significantly lower expression in *CsCYP82D102* overexpression plants relative to WT plants, both before and after infection. In contrast, the basal expression of *PR5*, a gene responsive to JA, was notably higher in the overexpression lines compared to WT plants. Following PM infection, both *PR3* and *PR4* expression substantially increased, but with lower levels detected in the overexpression plants. These findings suggest that the upregulation of *CsCYP82D102* expression in cucumber leads increases the expression of *PR5*, a key component involved in SA-mediated defense signal transduction. Additionally, the enhanced resistance to powdery mildew observed in the transgenic cucumber plants may be attributed to the upregulation of these genes.

To further elucidate the impact of *CsCYP82D102* on signaling transduction, the expression of key regulators within the SA/MeJA signaling pathways was examined. Among them, *LOX2* (lipoxygenase 2), which is involved in JA biosynthesis and signaling pathway, exhibited intensified induction in the overexpression plants following infection. *JAR1* (jasmonate resistant 1), responsible for JA conjugation to isoleucine and the production of biologically active jasmonate-isoleucine (JA-Ile), displayed significantly higher expression levels in the overexpression plants after infection. Moreover, the expression of defense-marker genes, such as *CAT2* and *APX*, was also modulated in PM-infected plants.

## DISCUSSION

P450 is the third-largest gene family involved in plant metabolism and the largest enzyme family. P450 genes account for approximately 1% of plant protein-coding genes, which is inferior to genes coding for transcriptional regulatory proteins (*Nelson & Werck-Reichhart, 2011*). P450 monooxygenase superfamily plays an important role in plant metabolism. Several studies have revealed the functions of P450 family genes. *Wang et al. (2022b)* screened 221 P450 genes from cucumbers. In contrast, the chosen cucumber genome (Chinese Long v2) and HMM profile (PF00067) exhibited slight differences. *Wang et al. (2021)* only made an integrated analysis of mobility direction and mRNA abundance of 15 mobile-mRNA-coding CsaP450 genes. In contrast, the P450 in cucumber, especially the CYP82 subfamily, was analyzed in detail. To investigate the evolutionary relationship between cucumber P450 genes, a phylogenetic tree was constructed using P450proteins from cucumber and Arabidopsis (Fig. S2). P450 proteins were classified into A-type (52.7%, 87/165) and non-A type (47.3%, 78/165). According to the criteria of phylogeny and homology, the P450 gene family was further divided into 40 families and eight clusters. Among these, the most extensive A-type gene family of cucumbers is CYP71, with 87 members. Studies have shown that the most prominent P450 gene family in most plants is CYP71, which contains more than half of the P450 genes and has rich and diverse functions, which is closely related to the metabolism of aromatic and aliphatic amino acid derivatives, some triterpenoid derivatives, fatty acids, alkaloids, and hormone precursors; the largest non-A type family in cucumber is CYP85, which is composed of nine gene families: CYP86, CYP707, CYP722, CYP716, CYP718, CYP87, CYP724, CYP85, and CYP90. It is worth mentioning that the CYP93, CYP701, CYP703, CYP51, CYP724,

CYP718, and CYP710 families are all composed of single cucumber genes, suggesting that each gene has a unique, highly conserved function. In particular, the CYP51 family is an ancient and conservative family, with only one member in the study of gene families of all species to date. CYP51G and CYP710A encode 14α-demethylase and sterol 22-desaturase, respectively, participating in sterol biosynthesis. In a survey of CYP703 family members, the Arabidopsis mutant *cyp703a2* showed the phenotype of pollen development retardation and partial male sterility (*Mizutani & Ohta, 2010*). In addition, compared to the A-type P450 family, the non-A-type P450 gene family has a wide range of species and complex changes. The non-A-type P450 gene family may be evolutionarily older than the A-type P450 family, and the time for gene replication and rearrangement may be longer. This leads to a more diverse composition than that of the A-type P450. The genome size of cucumber is 367 Mb, which is 2.7 times that of Arabidopsis (135 Mb), but the number of P450 genes in cucumber is 68.8% of that in Arabidopsis, suggesting that some P450 genes were lost during the evolution of cucumber (*Jiu et al., 2020*). The distribution of P450 in *Arabidopsis* and cucumber showed a similar trend. A-type P450 is plant-specific and is characterized by a remarkably consonant intron and a simple organization. Relevant A-type P450 genes are closely clustered in the genome; there are multiple genes on a minor chromosome segment (for instance, CYP71 subfamily genes). A-type P450 enzymes may play a complex role in plant-specific biochemical pathways, while non-A type P450 genes form several distinct branches, each characterized by numerous introns. In cucumber, the A-type P450 genes account for more than non-A-type, which is similar to that of *Arabidopsis* (63%, 167 out of 264) (Fig. S2), indicating that cucumber has evolved a great number of P450s for the biosynthesis of diverse metabolites, just as has *Arabidopsis* (*Ma et al., 2014*). On this basis, 12 CYP82 genes were discerned and delineated by referencing the cucumber genome database. These findings indicated proximity to the numbers found in widely cultivated plants like popular (10) (*Minerdi, Savoi & Sabbatini, 2023*) and tomato (nine) (*Vasav & Barvkar, 2019*), while falling below the counts observed in soybean (24) (*Guttikonda et al., 2010*) and grape (34) (*Jiu et al., 2020*; *Minerdi, Savoi & Sabbatini, 2023*). Conversely, the count exceeded those in *A. thaliana* (five) and various other plants, including Maize (*Frey et al., 1995*; *Li & Wei, 2020*), Rice (*Wang et al., 2022a*), and Moss (0) (*Katsumata et al., 2011*). Additionally, it is noteworthy that this outcome aligns with the genome size of different species, thus demonstrating a positive correlation between the quantity of CYP82 family genes and the genome size of the respective species. The CYP82 family makes cotton strongly resistant to disease and stress, and it is speculated that it has a similar role in cucumbers (*Wei & Chen, 2018*). The CYP93 family is involved in the biosynthesis of soybean flavonoids, with 13 members in soybean and seven in maize. Like Arabidopsis and rice, cucumber has fewer members of the CYP93 family, further confirming the specificity of the P450 gene family.

There are relatively conserved motifs in the cucumber P450 gene, and a heme-binding domain in each CsCYP450. CsCYP450 proteins from the same subfamily have similar motifs, indicating high function similarity, which was further confirmed through gene structure analysis. The increase and decrease of introns are common phenomena during plant evolution, which enrich the complexity of gene structure (*Wei & Chen, 2018*). There
is an excellent correlation between phylogeny and intron conservation among members of the CYP82 subfamily (*Jiu et al., 2020*).

Gene duplication offers a fundamental and plentiful source of genetic material crucial for plant evolutionary processes. Cucumber has experienced many tandem replications and fragment replications, and a large number of replication genes will mutate randomly, resulting in expression and functional differentiation under different selection pressures. This also explains why replicative genes non-functionalization, sub functionalization, and informational functionalization have compared with their original function. Overall, gene replication is a vital driving force in plant evolution. Selection pressure can promote the retention of replication genes with new functions. For example, MADS-box transcription factors are involved in the structural evolution of floral organs, dioxygenase genes are involved in pigment variation incarnation, and so on. The CYP71 family accounts for more than half of the gene family in higher plants, and this burst of gene replication is likely to contribute to the adaptation to specific ecological niches and speciation. During the evolution of cucumber, a rapid increase in CsCYP82s as due to the main mechanism of tandem replication, similar to the mechanism of grape expansion (*Nelson & Werck-Reichhart, 2011*). The unequal exchange of alleles mainly causes tandem duplication and occurs in all chromosomes except in chromosome 7 in cucumbers. Functional differentiation occurred in cucumbers during evolution. Certain P450 families in grape, for example, CYP76 and CYP82, experienced numerous duplications, resulting in extensive clusters containing similar sequences. Examining gene expression unveiled remarkably distinct expression patterns, frequently shared by the genes located within substantial physical clusters (*Jiu et al., 2020*).

To further explore the evolution and origin of the *CsCYP450* gene family in cucumber, the collinearity relationship between Arabidopsis, cucumber and melon was analyzed at the genomic level. According to the collinear analysis, in addition to one-to-one matching, there is also many-to-one matching. These results suggest that the functional differentiation of these genes may have occurred in cucumbers and melons during evolution.

In the process of differentiation, development, and growth, plants must integrate environmental and developmental signals into different tissues to regulate gene expression. As an important component of transcriptional regulation, cis-elements of the promoter are involved in regulating networks involved in many biological processes. Our study found many repetitive areas in the promoter of *CsCYP82s*, which conformed to the results of previous studies (*Wen et al., 2020*; *Zhang et al., 2020*). The identified regulatory elements were divided into four groups, excluding specific and general elements, and including elements responsive to growth and development, hormones, pressure, and light. In the current study, some cis-regulatory elements (ACE, Box 4, MRE, *etc.*) were found in the promoter of CsCYP82s, which are required for light-driven transcriptional regulation, implying that these genes played an important role in the light-responsive process (*Li et al., 2022*).

These results demonstrate that P450 is involved in many biochemical pathways and is important in plant growth and development. CYP82 exists only in dicotyledons and has
been confirmed to perform critical biological functions in soybean and cotton. For instance, CYP82 in soybean is expressed in the stems, leaves, and roots, with the highest expression level, and SA possesses a strong induction effect. The results of this study were consistent with this. CsCYP82 subfamily genes, Csa3G852630 and Csa3G852560 were respectively up-regulated in root and leaf. In addition, the Csa3G852630 also contains specific promoter motifs, auxin-responsive elements, and TC-rich repents (defense and stress-responsive element). These motifs are thus implicated in gene expression mediated by auxin and in countering oxidative stress in plants. Studies have shown that phenomena and products are more common in the biological functions of gene families. For example, the *IFS1* gene in soybean is preferentially expressed in roots and seeds- but not in stems, whereas the expression level of genes involved in fat hydroxylase is minimal.

The difference in the pattern was that the expression level of Csa5G606840, Csa7G198310 and Csa6G641590 reached the highest at 12 h, while the expression level of other genes reached the highest at 6 h after SA treatment. The *CYP749A45* (Csa5G606840) gene was up regulated in roots, but not in stems and leaves with unknown cucumber functions. This may be because the position and function of P450 in the biochemical pathway are different; therefore, it is not surprising that the expression level is significantly different. SA has been identified as a complex regulator of plant disease resistance. In addition, the CsCYP450 promoter also contains cis-regulatory elements related to hormonal responses and biological stress. Whether these findings indicate that *CsCYP450* plays an intermediary role in activating the intracellular SA signaling pathway to regulate cucumber resistance needs to be further verified.

Some researchers have documented the roles of CYP82 genes in plant disease resistance (*Sun et al., 2014*; *Xia et al., 2023*). In this investigation, rapid and robust *CsCYP82D102* was evident in cucumber plants after SA treatment, and the overexpression of *CsCYP82D102* in cucumber resulted in improved resistance against powdery mildew pathogens. Subcellular localization showed that the gene was located in the endoplasmic reticulum, consistent with the previous prediction. POD, SOD, and CAT are vital components of the plant's antioxidant system, and their activities reflect the plant's response to external environmental stress (*Mukarram et al., 2023*; *Shah & Nahakpam, 2012*; *Zhang et al., 2022*). These enzymes work synergistically to maintain the levels of free radicals within plants, thereby preventing physiological and biochemical changes associated with stress-induced free radicals. In this study, it was observed that the activities of CAT, POD, and SOD were significantly higher in transgenic cucumber plants compared to wild-type plants when exposed to powdery mildew pathogens.

Additionally, leaves of *CsCYP82D102* overexpressing plants exhibited lower intensity of staining with DAB and NBT compared to WT plants post PM infection. The observed phenomenon was absent in transgenic plants subjected to RNA interference (RNAi). Cucumber harbors a total of 12 members belonging to the CYP82 family. The sole silencing of *CsCYP82D102* cannot induce a change in its function, suggesting the presence of a phenomenon associated with functional redundancy. Transcriptome analysis of phytohormone-related genes showed the activation of *PR5*, associated with SA signaling, while the upregulation of JA signaling genes (*LOX2*, *JAR1*) in the transgenic plants

indicated the activation of the JA signaling pathways mediated by *CsCYP82D102*. Additionally, modulation in the expression of defense-marker genes, including *CAT2* and *APX*, was observed upon PM infection. However, further research is required to elucidate the underlying mechanism(s) involved. The limitation of our study lies in the narrow focus on verifying the powdery mildew function of the *CsCYP82D102* gene alone. Future investigations should expand to explore its potential functions in other bacteria. The study employed RNAi interference technology, which has limitations. To enhance precision, future research should incorporate CRISPR technology for gene knockout, enabling a more accurate analysis of *CsCYP82D102* function. Additionally, since various members of the CYP82 family participate in diverse metabolic pathways, further characterization of their biochemical functions is warranted. To confirm the functions of the CYP450 gene family, future research could entail comprehensive functional assays, genetic manipulation studies, and comparative analyses across different organisms. These approaches would provide deeper insights into the roles of CYP450 genes in various biological processes.

## CONCLUSIONS

In summary, our investigation identified 12 CsCYP82 genes in cucumber, revealing their diverse roles in development, light and hormonal responses, and stress conditions. Analysis of the phylogenetic tree and motifs revealed that the CYP82 proteins are evolutionarily conserved. In addition, the function of the *CsCYP82D102* gene was verified. After PM infection, the overexpressed transgenic plants exhibited stronger disease resistance, associated with the upregulated expression of genes involved in the SA and MeJA pathway. In conclusion, this study lays a foundation for further exploring the biological function of the CsCYP82 genes in cucumber. Future research should delve deeper into specific stress responses, guiding potential applications for crop improvement.

## ACKNOWLEDGEMENTS

The cucumber varieties "9930" was kindly provided by Prof. Huang Sanwen (Institute of Vegetables and Flowers, Chinese Academy of Agricultural Sciences). We thank Editage Services for editing the English text of a draft of this manuscript.

### Funding

This work was supported by the Harbin Normal University Postgraduate Innovation Project (HSDBSCX2021-02) and the Natural Science Foundation of Heilongjiang Province (LH2021C052). The funders had no role in study design, data collection and analysis, decision to publish, or preparation of the manuscript.

### Grant Disclosures

The following grant information was disclosed by the authors:
Harbin Normal University Postgraduate Innovation: HSDBSCX2021-02.
Natural Science Foundation of Heilongjiang: LH2021C052.

## Competing Interests

The authors declare that they have no competing interests.

## Author Contributions

- Hongyu Wang conceived and designed the experiments, performed the experiments, analyzed the data, prepared figures and/or tables, and approved the final draft.
- Pengfei Li performed the experiments, prepared figures and/or tables, and approved the final draft.
- Yu Wang performed the experiments, analyzed the data, prepared figures and/or tables, and approved the final draft.
- Chunyu Chi analyzed the data, prepared figures and/or tables, and approved the final draft.
- Guohua Ding conceived and designed the experiments, analyzed the data, authored or reviewed drafts of the article, and approved the final draft.

## Data Availability

The sequences are available at NCBI: PRJNA80169; PRJNA321023.

## Supplemental Information

Supplemental information for this article can be found online at http://dx.doi.org/10.7717/peerj.17162#supplemental-information.

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
