# Peer review of "Genome-wide identification of the CYP82 gene family in cucumber and functional characterization of CsCYP82D102 in regulating resistance to powdery mildew"

_PeerJ, doi:10.7717/peerj.17162_

## Round 0.1 · original submission · Minor Revisions

Please revise your manuscript according to the comments from the reviewers.

Reviewer 2 has suggested that you cite specific references. You are welcome to add it/them if you believe they are relevant. However, you are not required to include these citations, and if you do not include them, this will not influence my decision.

**Language Note:** PeerJ staff have identified that the English language needs to be improved. When you prepare your next revision, please either (i) have a colleague who is proficient in English and familiar with the subject matter review your manuscript, or (ii) contact a professional editing service to review your manuscript. PeerJ can provide language editing services - you can contact us at copyediting@peerj.com for pricing (be sure to provide your manuscript number and title). – PeerJ Staff

Reviewer 1 ·

Basic reporting

The manuscript is clear, unambiguous, and technically correct. The introduction, background issues, and references cited are sufficiently reported. The article structure is acceptable. Raw data have been checked, and they are consistent. The figure's resolution is not good, and all the figure's resolution MUST be improved.

Experimental design

Well-designed. Methods are consistent and well described.

Validity of the findings

No comments.

Additional comments

- The resolution of all the figures is terrible. The authors must improve the resolution.
- Figure S2 is not cited in the manuscript.
- A fluent speaker should proofread the manuscript. Please see some suggested English and punctuation corrections in the attached file.
- In the abstract, there is a lack of cis-acting element prediction results. The abstract does not provide a good outline for the study.
- What is the limitation of this study? What can be further done to confirm the functions of (CYP450) gene family can be discussed.
- How was the heat map constructed?
- Arrange the keywords in alphabetical order.
- All gene names must be in italics.
- All the scientific names in the references must be in italics.

Annotated reviews are not available for download in order to protect the identity of reviewers who chose to remain anonymous.

Reviewer 2 ·

Basic reporting

Literature references, and sufficient field background/context provided.
Self-contained with relevant results to hypotheses.

Experimental design

The research question is well-defined, relevant & meaningful.
Methods described with sufficient detail & information to replicate.

Validity of the findings

No comment

Additional comments

Recommendations:
The manuscript entitled as “Genome-wide identification of the CYP82 gene family in cucumber and functional characterization of CYP82D102 in regulating resistance to powdery mildew” can be accepted after substantial changes. It is good work but it needs few additional points.
The subject of the study is very interesting and topical, with scientific and practical importance. The data presented are of more than national interest.
Title is good.
The abstract section is very hard to understand so please carefully revise it with the following information; background information, aim, objectives, methodology, objectives, results and conclusion information. The author should mention the most prominent results in % in abstract.
Kindly use the scientific name of the cucumber throughout the manuscript.
Arrange the keywords alphabetically and start new word with small letter.
The introduction articulated very well but it needs a little bit improvement, for instance the CYP82 gene family should be introduced better and a clear working hypothesis should be introduced. Some of following literature should be consulted and cited in introduction and discussion section.

https://doi.org/10.1038/s41598-023-48947-z
https://doi.org/10.1186/s12870-023-04647-4
https://doi.org/10.1016/j.jhazmat.2023.132955

Why CYP82 gene family and CYP82D102 was chosen?
The authors should elaborate the Materials and Methods part more in detail.
Authors estimated the genes expression via qRT-PCR. But I have seen no figure of gene amplification in the manuscript. Authors must add a raw gel/blot image of gene amplification with proper PCR protocol and conditions before qRT-PCR.
Mention the specs of qRT-PCR instruments.
The qRT-PCR protocol needs to be elaborate well.
The protocol is too descriptive, especially for qPCR. How the author checks the specificity/efficiency of primers? Which kind of solution mixture was used for qPCR? I suggest including the details of the methodology of respective assays.
Tables and figures should be clear. I would recommend avoiding abbreviations in titles or add abbreviations meaning in figure legends. Tables should be arranged on portrait page. Figures and tables should be understood without reference to the text. Therefore, all necessary information should be included in the table/figure itself or the corresponding legend, for example, the definition of all abbreviations or a brief description of the treatments and the statistical analysis. Legends of figures and tables in the manuscript should be extended accordingly.
Discussion is good enough to support the results.
The list of references should be carefully checked and corrected. Some references do not seem to correspond to the text where they are cited and appear to have been randomly selected.
Please add some comprehensive conclusions in the end. It looks like a general description. It should be more concise and attractive. Also add some future recommendations.
Please solve all the abbreviations, even if it is evident and then apply them consequently, please check all the text.
Revise italics throughout the manuscript
English should be improved. The manuscript requires a thorough proofread by a person who is fluent in English.

·

Basic reporting

it is all ok!

Experimental design

it is all ok!

Validity of the findings

it is all ok!

Additional comments

The research content of this paper is substantial, the analysis is proper, and the logic is clear. At the same time, the transgenic experiment of cucumber is also carried out and the preliminary experimental results are obtained. It is a good paper, and it is recommended to accept it directly after minor revision. The comments are as follows:
1. The pixel of the picture, as shown in Figure 1, needs to be increased.
2. The legend in Figure 2 can be more detailed.

---

## Round 0.2 · accepted · Accept

Your revised version of the manuscript is accepted for publication.

Reviewer 1 ·

Basic reporting

I appreciate the authors' effort in revising the manuscript satisfactorily.

Experimental design

I appreciate the authors' effort in revising the manuscript satisfactorily.

Validity of the findings

I appreciate the authors' effort in revising the manuscript satisfactorily.

Additional comments

I appreciate the authors' effort in revising the manuscript satisfactorily. The current version is suitable for publication in PeerJ.

Reviewer 2 ·

Basic reporting

Self-contained with relevant results to hypotheses.

Experimental design

Research question well defined, relevant & meaningful. It is stated how research fills an identified knowledge gap.

Validity of the findings

No comment

·

Basic reporting

yes

Experimental design

yes

Validity of the findings

yes

Additional comments

yes